# The Crystal Structure of the *Plasmodium falciparum* PdxK Provides an Experimental Model for Pro-Drug Activation

**Kai Gao** [1] ⓘ, **Wenjia Wang** [1] ⓘ, **Thales Kronenberger** [2,3] ⓘ, **Carsten Wrenger** [3] **and Matthew R. Groves** [1,*]

1   Drug Design XB20, Groningen Research Institute of Pharmacy, University of Groningen, 9700 AD Groningen, The Netherlands; k.gao@rug.nl (K.G.); w.wang@rug.nl (W.W.)
2   Department of Medical Oncology & Pneumology, Internal Medicine VIII, University Hospital Tübingen, 72076 Tübingen, Germany; kronenberger7@gmail.com
3   Unit for Drug Discovery, Department of Parasitology, Institute of Biomedical Sciences, University of São Paulo, 05508-000 São Paulo, Brazil; cwrenger@icb.usp.br
*   Correspondence: m.r.groves@rug.nl; Tel.: +31-(0)50-363-3305

**Abstract:** Pyridoxine/pyridoxal kinase (PdxK), belongs to the ribokinase family and is involved in the vitamin B6 salvage pathway by phosphorylating 5-pyridoxal (PL) into an active form. In the human malaria parasite, *Plasmodium falciparum*, *Pf*PdxK functions to salvage vitamin B6 from both itself and its host. Here, we report the crystal structure of *Pf*PdxK from *P. falciparum* in complex with a non-hydrolyzable ATP analog (AMP-PNP) and PL. As expected, the fold is retained and both AMP-PNP and PL occupy the same binding sites when compared to the human ortholog. However, our model allows us to identify a FIxxIIxL motif at the C terminus of the disordered repeat motif $(XNXH)_m$ that is implicated in binding the WD40 domain and may provide temporal control of *Pf*PdxK through an interaction with a E3 ligase complex. Furthermore, molecular docking approaches based on our model allow us to explain differential *Pf*PdxK phosphorylation and activation of a novel class of potent antimalarials (PT3, PT5 and PHME), providing a basis for further development of these compounds. Finally, the structure of *Pf*PdxK provides a high-quality model for a better understanding of vitamin B6 synthesis and salvage in the parasite.

**Keywords:** *Pf*PdxK; motif; PT3; PT5; PHME

## 1. Introduction

Pyridoxal 5′-phosphate (PLP) is the active form of vitamin B6, a cofactor for many enzymes involved in amino acid and sugar metabolism [1]. Many species possess both a *de novo* PLP synthetic pathway, as well as a salvage pathway, to take up vitamin B6 from nutrients. In the presence of ATP and $Mg^{2+}$, precursors of an active form of vitamin B6 (pyridoxine (PN), pyridoxal (PL), and pyridoxamine (PM)) are phosphorylated by PL kinase (PdxK) through transfer of a phosphate group from ATP to the 5′-hydroxyl group of PL [2,3]. PL kinase has been studied and purified from bacterial, plants, and mammalian sources and most organisms contain a single PL kinase, coded for by the PdxK gene [4,5]. The crystal structures of sheep brain and human PdxK have been previously published [6,7]. Additionally, *E. coli* PdxK structures have been determined both as native and binary complexes (with either Mg:ATP and PL) [8]. Despite a low sequence identity among these organisms, they share a highly similar structure and fold [9]. As vitamin salvage has long been thought to be a potential area for the development of novel antimalarials, the PdxK from *P. falciparum* has been studied by electron microscopy [10], and we have previously reported the crystallisation of *Pf*PdxK [11].

Unlike mammalian cells, *P. falciparum* has a functional vitamin B6 *de novo* biosynthetic pathway [12], which has been previously validated as a drug target [13]. The parasite also possesses an interconversion pathway reliant upon *Pf*PdxK, which has been examined for pro-drug discovery [14]. A strategy of impairing the growth of *P. falciparum* was suggested, in which pro-drugs are phosphorylated within the parasite to generate pyridoxyl-amino acid adducts that block PLP-dependent enzymes, inhibiting proliferation [15]. Three novel non-phosphorylated pyridoxyl-adducts compounds (PT3, PT5 and PHME) were tested in an anti-plasmodial assay, with results showing that PT3 can inhibit the proliferation of *P. falciparum* with an IC$_{50}$ of 14 μM [15]. Further development of this strategy requires high-resolution models of both the human and *plasmodial* PdxK, in order to guide improvements in specificity.

*Pf*PdxK is also rather unique amongst PdxKs, as malarial sp. PdxKs possess a highly degenerate internal motif (XNXH)$_m$ that shares similarities with motifs involved in targeting protein for degradation, thereby potentially providing temporal control of PdxK activity. This type of degenerate motif is common within the malarial genome [16] and little research has been performed on any advantage such degenerate motifs may convey to the parasite. Interestingly, while this degenerate loop was predicted to be non-structured, a second accompanying motif (FxIxxIL) found that the N-terminal to (XNXH)$_m$ may provide some insight into the molecular interactions between *Pf*PdxK (and other degenerate repeat containing proteins) and the molecules(s) likely to govern their degradation.

In this manuscript, we report the crystal structure of the PdxK:Mg:AMP-PNP:PL complex from *P. falciparum* determined by X-ray diffraction at a resolution of 2.15 Å. We show, as expected, that the overall fold is highly conserved with other members of the PdxK family and that the (XNXH)$_m$ motif is disordered. However, the (FIxxIIxL) motif is structured and provides a surface-exposed epitope that shares similarities with the EH1 motif that plays a role in recruiting WD40 domains. Molecular modelling also demonstrated the molecular rational for the improved behavior of PTME and PT5 over that of PT3.

## 2. Materials and Methods

### 2.1. Purification and Crystallization

The cloning, expression, buffer optimization and crystallization procedures of native *Pf*PdxK have been described previously [11]. Briefly, *Pf*PdxK was recombinantly expressed in *E. coli* Rosetta 2 (DE3) cells, followed by propagation in LB medium supplemented with 4 mM MgCl$_2$ at 37 °C, after reaching the OD$_{600}$ at 0.6, anhydrotetracycline was added for induction of protein expression and the culture was further incubated at 18 °C overnight until the pellets were harvested. Initial purification was performed using the affinity chromatography column, HisTrap FF (GE Healthcare). The protein was eluted with loading buffer supplemented with 300 mM imidazole, then the protein was pooled, concentrated and purified by size exclusion chromatography using column HiLoad 16/60 Superdex 75 (GE Healthcare). Protein purity was assessed by SDS-PAGE and apo crystals appeared in hanging-drop experiments performed at 20 °C using a crystallization buffer containing 0.1 M HEPES, pH 7.75, 0.2 M CaCl$_2$, 31%(*v/v*) PEG 400, 5% (v/v) glycerol. AMP-PNP and PL (each 5 mM) were added into the drops containing apo *Pf*PdxK crystals for soaking. The resulting crystals were then flash-cooled in liquid nitrogen with an additional 25% (v/v) glycerol as a cryo-protectant. All chemicals were purchased through VWR, Amsterdam, The Netherlands.

### 2.2. Data Collection and Structure Determination

X-ray diffraction data were collected at EMBL Hamburg and P11, the PETRA III synchrotron facility, Hamburg, Germany. Diffraction images were processed with XDS [17], scaled and merged with Aimless [18]. While the crystals displayed a variation in the recorded diffraction resolution from 3.4 Å to 2.1 Å, all *Pf*PdxK complex crystals were processed in space group P1211. An overview of the data collection and refinement statistics can be found in Table 1. The structures of the *Pf*PdxK/AMP-PNP complex were solved by molecular replacement with PHASER [19] using the coordinates of PdxK from sheep brain as a search model (PDB entry 1RFU [6]). The model was further improved by iterative

cycles of manual rebuilding via Coot [20] and refinement using Refmac5 [21] to improve the electron density map. All refinement steps were carried out automatically, using software default values and applying non-crystallographic restraints.

**Table 1.** Data collection and refinement statistics.

| Data Set | *Pf*PdxK-AMP-PNP-PL |
|---|---|
| Wavelength | 0.97 Å (12.398 keV) |
| Resolution range | 19.33–2.15 (2.227–2.15) |
| Space group | P 1 21 1 |
| Unit cell | 52.703 62.004 93.712 90 94.988 90 |
| Total reflections | 81052 (8023) |
| Unique reflections | 32284 (3159) |
| Multiplicity | 2.5 (2.6) |
| Completeness (%) | 97.87 (97.17) |
| Mean I/sigma(I) | 15.57 (2.69) |
| Wilson B-factor | 39.31 |
| R-merge | 3.7 (38.61) |
| R-meas | 4.7 (49.31) |
| CC 1/2 | 0.99 (0.88) |
| CC * | 1 (0.967) |
| Reflections used in refinement | 32276 (3159) |
| Reflections used for R-free | 1627 (165) |
| R-work | 0.2094 (0.3002) |
| R-free | 0.2718 (0.3691) |
| Number of non-hydrogen atoms | 4885 |
| macromolecules | 4727 |
| ligands | 89 |
| solvent | 69 |
| Protein residues | 591 |
| RMS(bonds) | 0.008 |
| RMS(angles) | 0.95 |
| Ramachandran favored (%) | 95.62 |
| Ramachandran allowed (%) | 4.20 |
| Ramachandran outliers (%) | 0.18 |
| Rotamer outliers (%) | 0.00 |
| Clashscore | 8.69 |
| Average B-factor | 59.75 |
| macromolecules | 59.58 |
| ligands | 73.01 |
| solvent | 54.41 |

The stereochemical properties of the final models were analyzed with the Molprobity server (http://molprobity.biochem.duke.edu). At least 95% of the residues are found in the favoured region of the Ramachandran plot. Protein–PL interactions were analyzed with the scorpion server (http://www.desertsci.com/). Structural alignments/analysis and all structural figures were prepared with PyMOL (Graphic System, Schrodinger, LLC). The structure has been deposited with the PDB under the accession code 6SU9.

*2.3. Molecular Docking*

Small-molecule docking of *Pf*PdxK with PT3, PT5 and PMME were performed by rigid-docking using Swiss dock [22] (http://www.swissdock.ch/docking), based on EADock. The coordinate file *Pf*PdxK was edited to remove all water molecules and PL, with AMP-PNP maintained in position to provide a realistic model of ATP binding. Compounds were optimized using MM2 via Chemdraw. The ligand-binding site was constrained in a $10 \times 10 \times 10$ Å$^3$ cube with the protein at the center. Flexible side chains were not allowed. All rotatable single bonds were allowed to rotate within the ligand. Docking results were screened by Chimera.

## 3. Results

*3.1. PfPdxK Adopts a Highly Similar Structure to that Seen in Different Species*

Multiple sequences and structures alignment were generated from T-Coffee [23] and ENDscript 2.0 [24]. We selected three published crystals structures from human, sheep brain and *E. coli* as reference

PdxK structures. *Pf*PdxK consists of 493 amino acids, of which, only 54% overlaps with the sequences of other species (Figure 1). Amongst this 54%, only 30% of the sequence was conserved. The insertion from His107 to Tyr303 consists of repeated-motif MNXH or TNXH peptides, with no known homology within the PDB (Figure 1). Sequencing of the expression plasmid confirm the completeness of the protein, with no detectable degradation visible [11]. Unfortunately, no electron density was visible for the degenerate region and there is likely no defined structure of the repeated motif. Based on our data, it seems that it does not influence the structure of *Pf*PdxK.

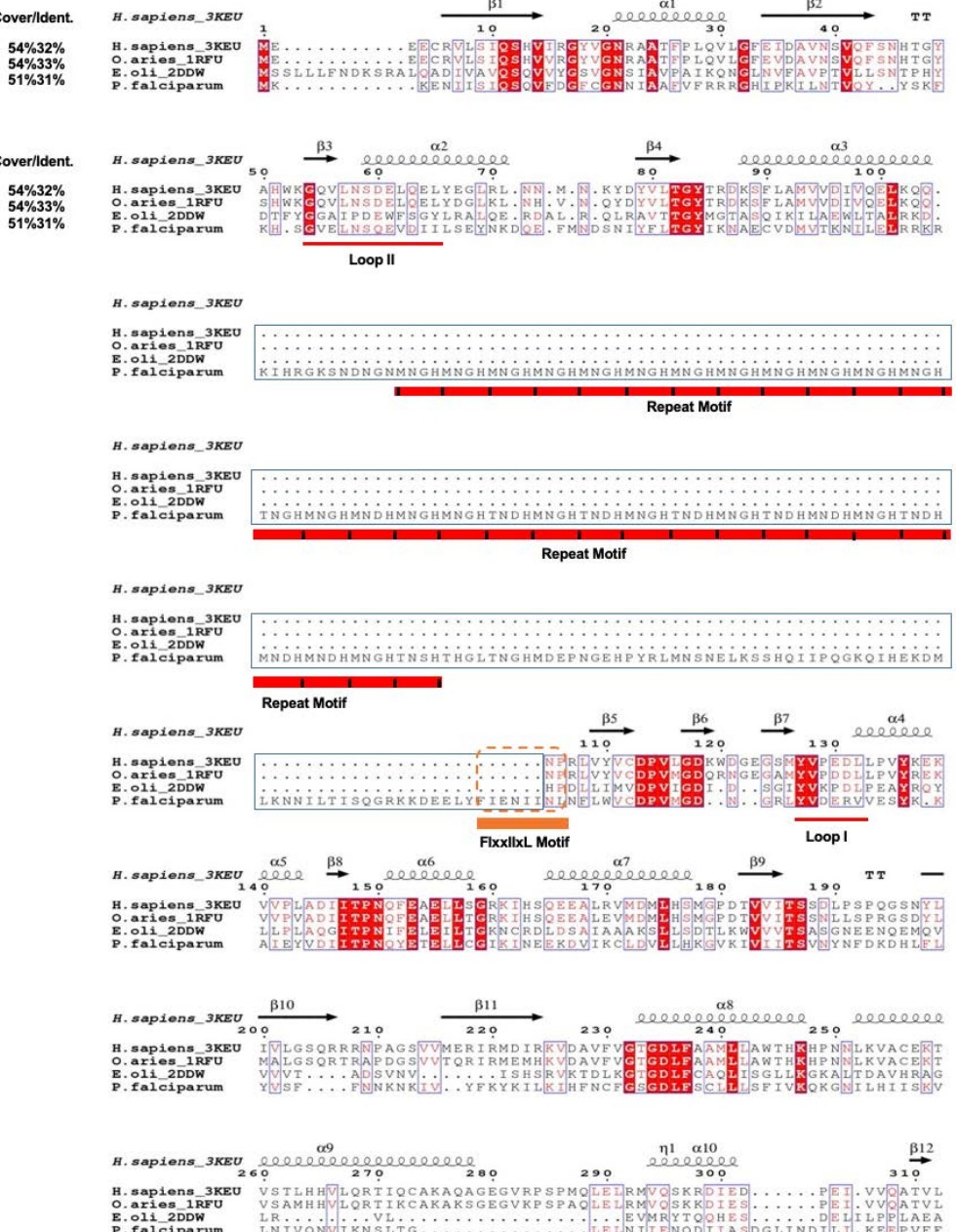

**Figure 1.** Multiple sequence alignment of *Pf*PdxK and the indicated homologs available within the PDB were generated using BLAST, T-Coffee and ENDscript 2.0. *Pf*PdxK exhibits a sequence identity of 30% over 54% of its sequence, with the remainder of the sequence present as an extended, high-degenerate insert consisting of multiple MNXH or TNXH repeats. Such repeats are frequently observed within the malarial genome (Figure S1 in Supplementary Materials), the Eh1-like motif **FIxxIIxL** was labeled in orange after the repeat motifs.

We performed a structural alignment of *Pf*PdxK against the human PdxK structure to uncover any conformational changes of *Pf*PdxK that may be driven by the internal repeat elements (Figure 2). As PdxK activity is retained in vivo, *Pf*PdxK unsurprisingly followed the basic structural organization of PdxK proteins, with the overall structure showing the same conformation as that typically seen within the ribokinase superfamily: each monomer consists of seven core β strands (β1–β7) with five parallel and one antiparallel strand, surrounded by nine α-helices (α1–α9) as indicated, with α2 slightly offset against α3. The active site is present as a shallow groove, flanked by conserved loop 1, with which ATP or AMP-PNP predominantly interacts. The 5-pyridoxal substrate binding site is equally conserved and almost completely buried in all structures, strongly suggesting a degree of rearrangement will be required during substrate loading and product dissociation [6].

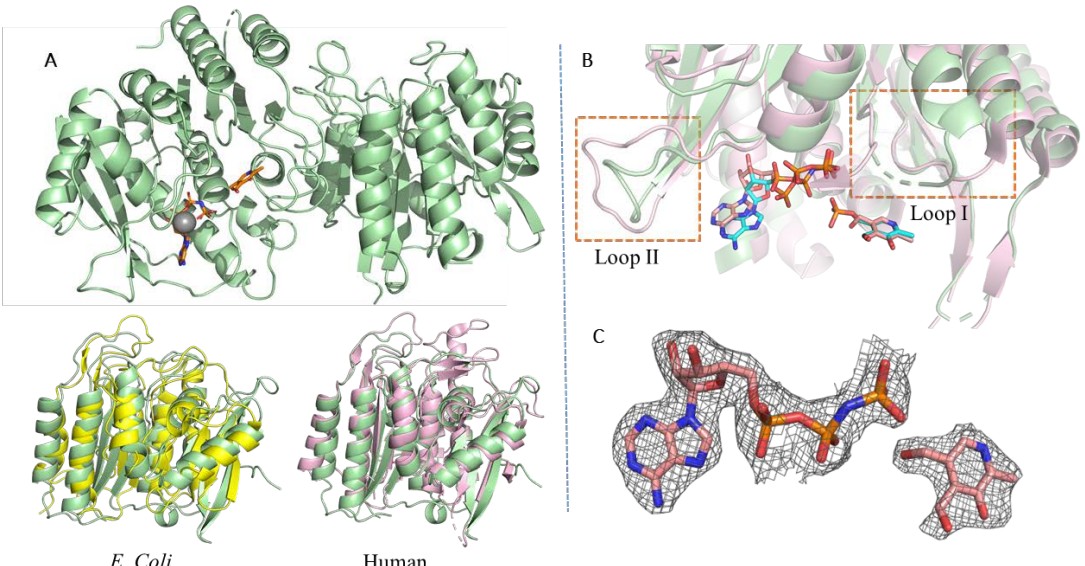

**Figure 2.** The structure of *Pf*PdxK loaded with the non-hydrolysable ATP analog (AMP-PNP) and PL (green). (**A**) Structure alignment with PdxK from Human (pink) and *E. coli* (yellow), respectively, demonstrates the conserved nature of the fold; (**B**) an alignment of AMP-PNP and PL in *Pf*PdxK (green) with the human complex (pink; 3KEU) with ADP and PLP shown in stick representation, showing the conserved nature of binding of both ATP and the substrate. (**C**) The *2Fo-Fc* electron density map for AMP-PNP and PL (black mesh) within the *Pf*PdxK structure (monomer A) contoured at 1σ. An omit map is for AMP-PNP and PL is shown in 3B and 3D.

### 3.2. PfPdxK in Complex with AMP-PNP and PL

The crystal structure of the *Pf*PdxK complex was determined at 2.15 Å (see Table 1 for crystallographic statistics) in space group *P*1211. Each asymmetric unit contains one molecule, and two monomers form a dimer by the two-fold crystallographic axis. Each monomer has an independent active site, with no contributions made from the opposing monomer. We found that the occupancy of AMP-PNP and PL is lower in monomer B than A, which is also visible in the higher B-factors of AMP-PNP and surrounding residues, indicating the *Pf*PdxK can also undergo conformational changes in the absence/presence of substrate, as seen in other PdxK enzymes [6]. A characteristic feature in the ribokinase superfamily structure is the flap or lid that cover the substrate, as well as the ATP phosphates group. Among the three structures of PL kinases, the flap is defined by loop I and loop II. During the binding of AMP-PNP, the flexible loop I around the active site was reported to trigger a major conformational change of the protein structure [6]. However, rotation of the flexible loop II is responsible for making close contact with and stabilizing the substrates.

In *Pf*PdxK, loop II adopts an extended conformation towards the buried active pocket and makes numerous interactions with PL. In one subunit, a conserved Cys18 on the flap binds to PL in a

non-covalent fashion. Ser11, His48 and Asp429 form hydrogen-bonding interactions with PL. In PLKs, the flap is in a more open position, and a hydrophobic residue replaces the cysteine. The movement of the flap probably makes a major contribution to the binding affinity between different PL substrates and this variation may explain why the orientation of PL in the binding site of our structure is slightly different from that seen in others. In the *Pf*PdxK complex, AMP-PNP and PL show a clear electron density signal in the active site *Pf*PdxK (Figure 2C). The adenine base makes two hydrogen bonds to Ile420, while the phosphate groups form multiple hydrogen bonds with the protein backbone both directly or via solvents (Figure 3). All residues that interact with the phosphate groups are highly conserved within the ribokinase family. The $Mg^{2+}$ ion is well coordinated by 4 highly ordered water molecules, Asp318 and the β phosphate of AMP-PNP.

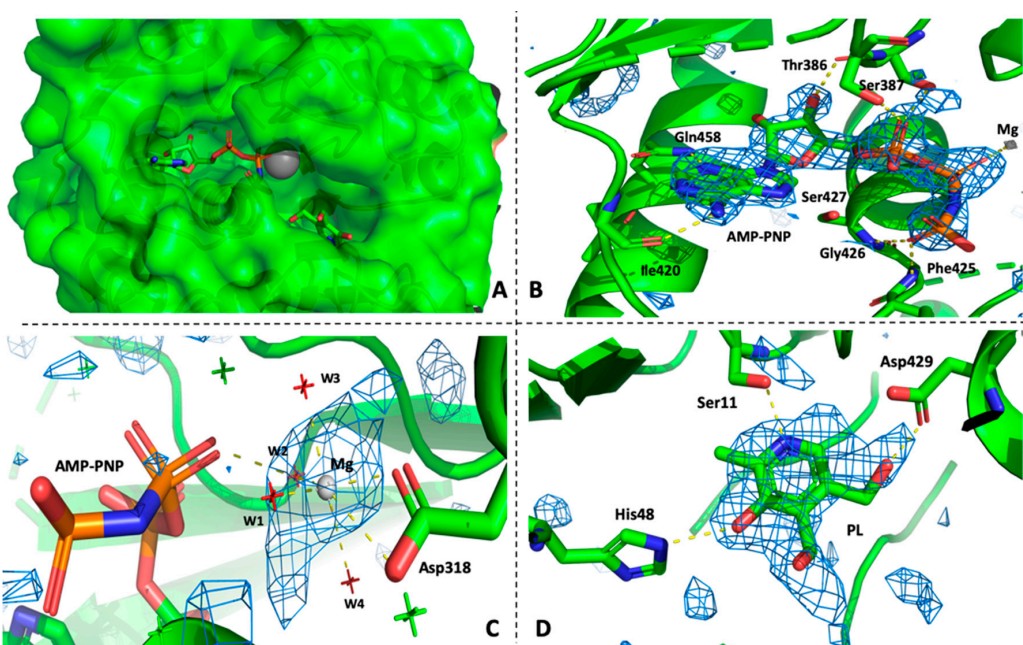

**Figure 3.** (**A**) Crystal structure of *Pf*PdxK in complex with AMP-PNP, $Mg^{2+}$ ion and PL, *Pf*PdxK is shown in the surface representation and three ligands located deep inside the pocket; The *Fo-Fc* omit map of AMP-PNP (**B**), $Mg^{2+}$ (**C**) and PL (**D**) within the *Pf*PdxK structure. Hydrogen bonding interactions of AMP-PNP, PL and $Mg^{2+}$ with *Pf*PdxK are shown as yellow dashed lines, the four red labeled water around $Mg^{2+}$ in (C) help to coordinate AMP-PNP and $Mg^{2+}$ ion in the pockets.

*3.3. The Surface-Exposed Conserved FIxxIIxL Motif may Function as a Signal for Degradation*

In the structure, we found that the degenerate sequence (located between α3 to α4 from His107 to Tyr303) was disordered. The equivalent region, connecting region α3 to α4, is a continuous loop or β sheet in other PdxKs (Figure 4A). The degenerate sequence contains numerous MNXH or TNXH repeat motifs that have no structural homologues in eukaryotes (Figure 1). However, it seems to be a frequent phenomenon in many plasmodial proteins (Figure S1). Further examination indicated not only that MNXH or TNXH repeats were often present, but that another motif (**FIxxIIxL**) conserved across apicomplexa is also often present at the C terminus of the degenerate inserts in other apicomplexan genes (Figure S2). The **FIxxIIxL** motif occupies a distinct conformation to that seen in the linker between α3 and α4 of human PdxK, suggesting that it does not function as a replacement of the connection between helices α3 and α4 (Figure 4C). A similar motif has been described previously (the Engrailed Homology1 (Eh1) motif) which has the consensus sequence **FxIxxIL**, of which only F is completely conserved. This motif functions by allowing the recruitment of other proteins. For example, the Eh1 motif was found to interact with the WD40 repeat domains of the Groucho/TLE co-repressor [25] (2CE8). Our model demonstrates not only that the **FIxxIIxL** motif is present as a surface-exposed epitope, but that it also adopts a similar conformation to that previously seen in the interaction of Eh1 with

WD40 repeats [25] (Figure 4A). The WD40 repeat consists of 40 amino acids, terminated with Trp-Asp (W-D) dipeptides. Repeated WD40 motifs act as a site for protein–protein interaction, and proteins containing WD40 repeats are known to serve as platform for the assembly of protein complexes, or mediators of temporal control among other proteins like G proteins, the TAFII transcription factor, or E3 ubiquitin ligase [26,27]. WD40 repeats usually assume a 7–8 bladed β-propeller fold. Proteins have been found with 4 to 16 repeated units, which also form a circular β-propeller structure [28]. In this structure, the Eh1 peptide binds over the mouth of the central channel of the TLE-WD40 domain (Figure 4B) and the central core of the Eh1 motif (from Ser2 to Leu7) forms a short amphipathic helix, following a very similar conformation to the **FIxxIIxL** motif in *Pf*PdxK.

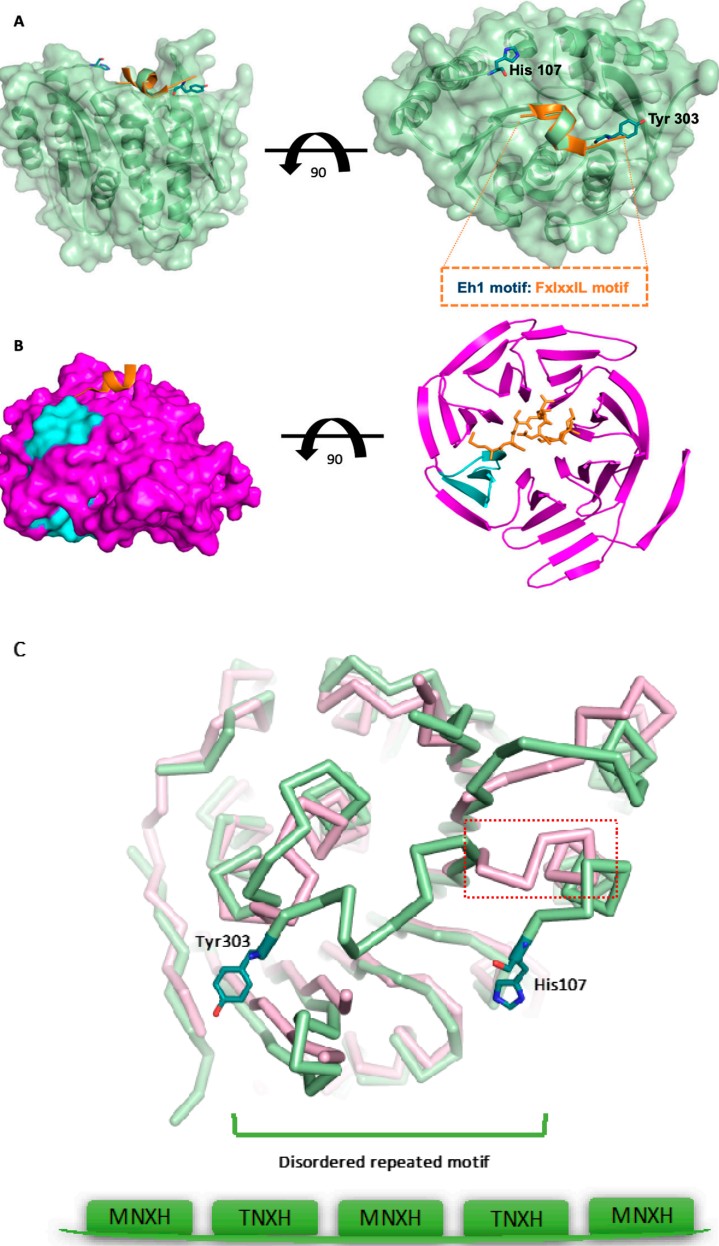

**Figure 4.** A surface-exposed epitope in *Pf*PdxK resembles the known Engrailed Homology1 (Eh1) motif. (**A**) View of an apicomplexian-conserved motif FIxxIIxL located on the surface of *Pf*PdxK, *Pf*PdxK is shown in green and an overlap of the Eh1 peptide is shown in orange. (**B**) Representation of the Eh1 motif (FxIxxIL) bound to the Groucho/TLE co-repressor with the WD40 domain shown in cyan. (**C**) Structure alignment of Human (pink) and *Pf*PdxK (green), demonstrating that the Eh1-like epitope does not directly replace the function of linking α3 to α4.

### 3.4. Molecular Docking Indicates Potential Binding Modes of PT3/PHME/PT5 as Substrates of PfPdxK

The novel non-phosphorylated pyridoxyl-adducts compounds PT3, PT5 and PHME were tested in an anti-*Plasmodial* assay, with results showing that PT3 can inhibit the proliferation of *P. falciparum* with IC$_{50}$ of 14 μM [15]. *Pf*PdxK was strongly implicated in the phosphorylation of PT3, PT5 and PHME into their active forms. As these three pro-drugs possessed distinct effects on *Plasmodial* cultures, we used our model to attempt to explain the differences in effect, based on their suitability for phosphorylation by *Pf*PdxK. To assess whether they were potential substrates for *Pf*PdxK and predict their binding models, we utilized the Swissdock online server based on grid dock to model the PT3/PHME/PT5 binding modes, respectively. As ATP is the source for PL phosphorylation, we prepared a single-molecule *Pf*PdxK with AMP-PNP bound as our molecular target protein, and non-phosphorylated pyridoxal-adducts PT3/PHME/PT5 as ligands to perform rigid docking (repeated three times independently).

According to a phosphorylation assay in vivo, PT3/PHME/PT5 were confirmed to be phosphorylated by *Pf*PdxK. All three were phosphorylated by the *Plasmodial* enzyme with a specific activity of 38, 20 and 64 nmol min$^{-1}$ m$^{-1}$, respectively [15]. We ranked potential docking on the lowest energy, selecting for positions immediately proximal to the AMP-PNP binding pocket that also possessed strong overlaps with the conserved PL-moiety (Figure 5). Our modeling suggests that the pyridoxal group of PHME and PT5 fit well onto the known *Pf*PdxK binding mode, but that the pyridoxal group of PT3 presented a potential steric clash with AMP-PNP, which might weaken the interaction of PT3 under the same conditions or require rearrangement of the ATP binding mode for phosphorylation to proceed. This calculation is consistent with the experiments that demonstrate PHME and PT5 possess better activities than PT3 as substrates of *Pf*PdxK and may provide the basis for further development of this pro-drug class.

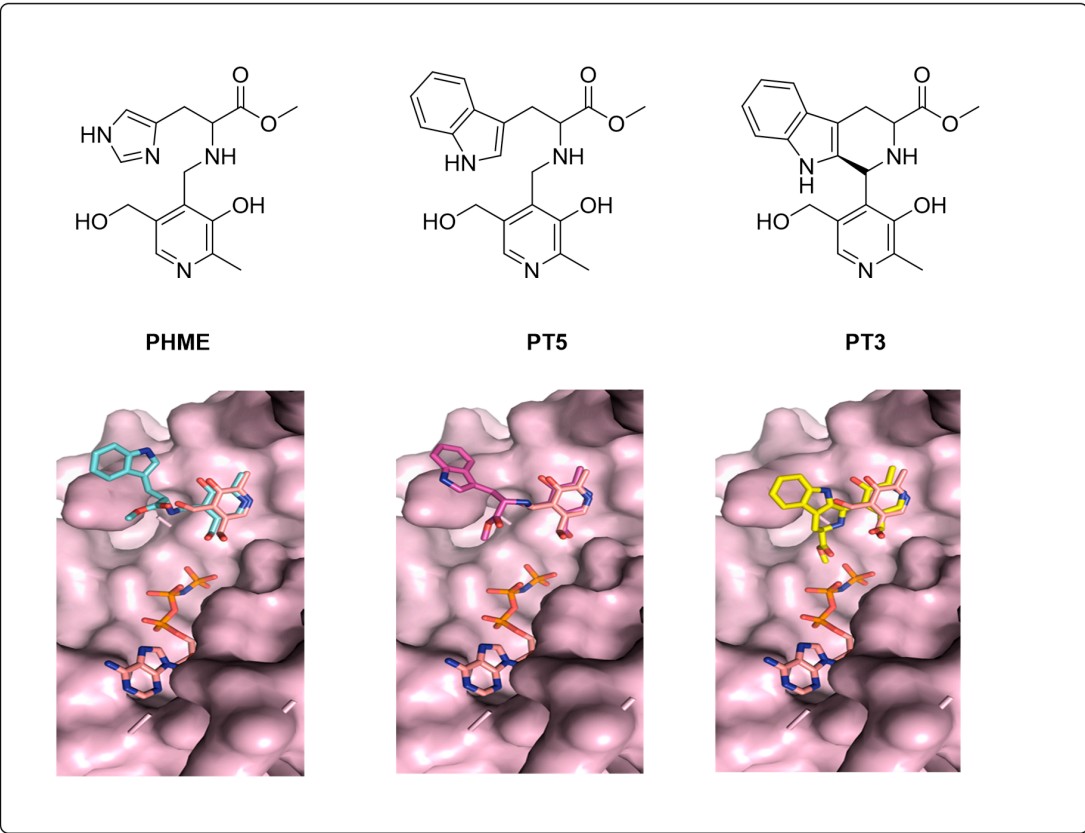

**Figure 5.** Molecular docking of PT3/PHME/ to *Pf*PdxK overlapped with AMP-PNP and PL.

## 4. Discussion

Based on the crystal diffraction data we obtained with AMP-PNP and PL, the overall structure of *Pf*PdxK shares a similar conformation with human and *E. coli* PdxK. The binding pocket for multiple substrates and ATP are conserved among the species as alignments indicated within the three species. Unfortunately, we were unable to detect any electron density signal from the repeating motif region, likely due to degradation during crystal growth. While the lack of a stable structure result is expected, we cannot exclude the possibility that this repeat region may become structured upon binding to a third, as yet unknown, partner. Similar repeats are found in other malarial sp. (Figure S1), suggesting a commonality of role for these repeats. As this repeat region is located at the opposite side of the protein to the substrate-binding pocket, it is unlikely that the motif itself would affect *Pf*PdxK function as a kinase. An examination of the solvent content of the crystal indicates that the truncated form of *Pf*PdxK presented here would possess a solvent content of 58%, in line with the solvent content often found in crystals, whereas the full-length *Pf*PdxK would possess a solvent content of 8%. Such a low value is extremely unlikely, suggesting that either a minor contaminant or auto-proteolysis is responsible for the removal of this degenerate sequence during the crystallisation process.

A potential explanation for this presence of the degenerate sequence in the wild-type enzyme is that these repeats provide a signal for post-translational control of *Pf*PdxK through an interaction with other proteins. Based upon a similarity between the **FIxxIIxL** motif found in *Pf*PdxK and the **FxIxxIL** motif found in the Engrailed Homology1 (Eh1) motif (both in sequence and in structure), we propose that one mechanism could be through the recruitment of a WD40 domain protein, that in turn, may recruit a E3 ligase to drive ubiquitin-mediated degradation of *Pf*PdxK and other plasmodial proteins containing similar pairings of degenerate MNXH or TNXK repeats with a C-terminal **FIxxIIxL** motif. This hypothesis is based on the report that the Eh1 motif may recruit WD40 proteins [29]. However, we have no data to support a direct interaction between a WD40 domain protein and *Pf*PdxK. Why the parasite, and other apicomplexians, require fine control of proteins, such as *Pf*PdxK, post-translationally remains an interesting open question.

While our model provides a molecular snapshot of ATP and PL binding to a pyridoxal kinase it should be borne in mind that interpretation of the electron density is always subjective. Our omit maps do indicate that PL is bound in the manner described, but we cannot exclude the possibility that some of the *Pf*PdxK molecules within the lattice do not have a bound PL, but are either empty or have a bound glycerol (present in the cryo-buffers used). However, our deposited model has good agreement in terms of PL orientation with other PL-bound pyridoxal kinases and the modeling of PT3, PT5 and PHME also suggest this is a binding site for PL-like molecules.

Additionally, we generated models by molecular docking to explain how *Pf*PdxK phosphorylates PT3, PT5 and PHME into the potential active anti-malaria drug. Our analysis shows that the predicted mode of binding of these molecules effectively explains the observed differences in their effect, with the differences in molecular properties in vivo accounted for by potential steric clashes with residues lining the binding pocket. However, it should be borne in mind that these results are obtained from in silico modeling and confirmation of the binding mode awaits experimental validation. The structure of the wild-type *Plasmodium* PdxK provides further insight into the molecular basis of active PT3/PHME/PT5 as pro-drugs. In all, the structure of *Pf*PdxK provides further basis for the understanding of *P. falciparum* parasite vitamin salvage.

**Supplementary Materials:** The following are available online at http://www.mdpi.com/2073-4352/9/10/534/s1, Figure S1: BLAST results of the MNXH or TNXH repeats domain alignment, commonly shown in plasmodium genome, Figure S2: Alignment of different resource plasmodial falciparum contain a common FIxxIIxL motif after the xNxH repeat motif domains, shows in the red square.

**Author Contributions:** Conceptualization, M.R.G. and C.W.; validation and analysis, K.G., W.W. and T.K.; data curation, K.G., W.W. and M.R.G.; writing—original draft preparation, all authors; writing—review and editing, all authors; visualization, K.G. and T.K.; supervision and project administration M.R.G.; funding acquisition, M.R.G. and C.W.

**Funding:** This research was funded by CSC Fellowships to KG and WW. The authors would like to also acknowledge the Fundação de Amparo à Pesquisa do Estado de São Paulo (FAPESP, grants 2014/03644-9 to TK, 2017/03966-4 and 2015/26722-8 to CW). This project has received funding from the European Union's Framework Program for Research and Innovation Horizon 2020 (2014-2020) under the Marie Skłodowska-Curie Grant Agreement No. 675555, Accelerated Early staGe drug discovery (AEGIS).

**Acknowledgments:** The authors would like to acknowledge access to synchrotron radiation at beamline X11, EMBL Hamburg and P11, PETRA III, DESY, Hamburg.

**Conflicts of Interest:** The authors declare no conflict of interest. The funders had no role in the design of the study; in the collection, analyses, or interpretation of data; in the writing of the manuscript, or in the decision to publish the results.

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
