# Peer review of "The Crystal Structure of the Plasmodium falciparum PdxK Provides an Experimental Model for Pro-Drug Activation"

_crystals, doi:10.3390/cryst9100534_

Round 1

Reviewer 1 Report

The manuscript by Gao et al., titled “The Crystal Structure of the Plasmodium falciparum PdxK provides an Experimental Model for Pro-drug activation” describes the high-resolution cocrystal structure of the Plasmodium falciparum Pyridoxine/pyridoxal kinase (PfPdxK) bound to its substrate 5-pyridoxal (PL) and non-hydrolyzable ATP analog AMP-PNP. The enzyme PfPdxK is involved in vitamin B6 salvage pathway in the malarial parasite and therefore presents an attractive target for the development of potent pro-drug based antimalarials. Using the structure, the authors performed structure-based docking of three known potential pro-drugs and proposed a model for their phosphorylation by the enzyme into active anti-malarial drugs.  Although similar structures of the enzyme from Human, E. coli and sheep brain are available and share the same fold, they possess differences in the structure and function, providing opportunities for development of antimalarials with more specificity. Overall, the manuscript is concise and well written and presents the structure and structure-based docking in a cohesive manner. The discussion section briefly but appropriately contextualizes this work by detailing the similarities and differences among this class of enzymes and provides future directions for the design of antimalarials with specificity against the parasite.

However, there are major issues with the structure that the authors need to address clearly, before the paper can be published (see below for comments). Since this work involves structure alone with no biochemical data, it is very important that the structural model is sound and free of large errors.

Major Comments/Suggestions:

The R-free value for the structure is much higher than normally accepted value of around 25 %. The values are around 30 % in the low-resolution shell and ~37 % in the highest-resolution shell, which are too high for an acceptable crystal structure. In addition, the difference between R-free and R-work is also greater than acceptable (5 %) value. While there could be multiple reasons for such high-value, such as twinning or NCS, it is surprising that the authors have not discussed these issues at all in the manuscript. This suggests that certain regions of structure are either not correctly built or well refined. Is this because of lack of density for the degenerate regions as the authors briefly mention, and therefore a less complete model? In addition, the % of Rotamer outliers and the RMS bonds and angles are also ion the higher side, further suggesting badly built and poorly refined model.   What is the CC-half value at the resolution (2.5 A) where the data was cut-off? The authors should include both the CC and CC-half values in Table 1, as it is standard in the field. Since phasing was done using molecular replacement (MR), the structural model built can be biased to the MR model used here. Without seeing the .pdb and .mtz files and/or the PDB validation report, one cannot objectively judge the accuracy of the built model under such high R-values.  Consider including a composite omit-map as Supplementary Figure that shows the quality of the structural model and any biases introduced during phasing and model building.

Minor comments/suggestions:

I recommend the authors to briefly describe the purification of the protein in the methods, instead of just mentioning ‘as described previously’ with reference. Plasmodium falciparum or falciparum should always be italicized. As the full name is mentioned in the abstract and introduction, use the abbreviated P. falciparum in all subsequent usage for consistency. Similarly, use coli and not E-coli (Line 103) Reference the figures either as Fig. or Figure through the text for consistency Line 22: correct “through a the E3 ligase complex” Line 101: Use PfPdxK and “PfPdxKs” and use “adopts a highly similar structure” – similar instead of ‘similarity’ Line 145: correct “E. col i” Figure 2A: Label the structures as ‘Human’ or “E. coli” in the figure panels, in addition to mentioning them in the legend Suggest showing the coordinated Mg2+ ion, along with the ligands (water) in one of the figure panels. Line 172: correct sp. Proteins Line 202: delete the second use of ‘potential’ in the title. Line 210: correct sources to ‘source’ Line 235: correct ‘kit’ to ‘it’ Line 235: delete ‘does’ Line 248: change ‘wide-type’ to wild-type

Reviewer 2 Report

My review report is attached as PDF file.

Reviewer 3 Report

Judging from the merging and refinement statistics, the model presented in this study can not be published in the current form:

The Rwork/Rfree gap of 9% and therefore too large, while Rfree should be below 30% for the reported resolution of 2.5A. Please use CC1/2 instead of Rmerge for resolution estimation. While the merging statistics look unsuspicious (except a low completeness in the high-resolution shell), the geometry has to be improved. Ramachandran statistics are bad (only 91.27% and 1.7% outliers) and there are too many rotamer outliers including a rather high clashscore.

I recommend the authors to thoroughly manually curate the structure as the model taken for MR (1RFU) already suffers from bad model geometries. In addition the authors could make use of e.g. the PDB-redo server (https://pdb-redo.eu) for model improvement. 

In addition to that, the discussion is rather short, lacks references and should therefore be improved.

I have outlined some minor comments below.

Line 116: In my version, the FIxxIIxL motif is labeled in orange.

Line 133: Citation? In the prior study, a HiLoad 16/60 Superdex 75 column was used for size-exclusion chromatography of PfPdxK. This column will not allow resolving a 120kDa dimer. Maybe I missed it but the size-exclusion chromatogram is not shown in the study cited.

Line 136: Conformational

Line 139: Which phosphate? If all, then change to phosphates.

Line 143: A better labeling of the figure would make it easier to immediately get the key features of the model. Where is the FIxxIIxL motif?

Line 145: E. coli

Line 149/158: A figure showing the unbiased Fobs–Fcalc difference map would be helpful here in addition to the 2Fobs-Fcalc map. 

Line 161: Am I missing something here? There is no Mg2+ ion shown in the structure, neither are water molecules. Why is the g-phosphate not involved in the octahedral coordination of Mg2+?

Line 163: The figure could be more thoroughly labeled. E.g. secondary structures containing residues that coordinate the ligands, as well as N- and C-termini.

Line 195: Figure 4 has a low resolution in my version. It should be improved.

Line 223: rearrangement

Line 235: ‘…, it is unlikely that the motif itself would affect…’

Line 243: Where has this been shown? Citation.

Round 2

Reviewer 1 Report

In the revised version of the manuscript, the authors have addressed the minor suggestions to the text and figures. However, they have not adequately and clearly addressed the major issues raised by all the reviewers related to the structure statistics.The statistics now look much better, but what is surprising is that the resolution has also improved quite significantly (2.15 A from 2.5 A before). This is a big difference in resolution and the authors did not make any attempt to explain how they achieved this, along with the vastly improved R-values, which is counter intuitive. In addition, although they mention 99 % completeness in the cover letter, it is only 97.15 % (close, but not the same) in the Data statistics Table. What is also perplexing to me is why some of the other statistics, mainly R-merge and I/sigma(I) have remained unchanged. The major issues need to be very carefully addressed and it is important to explain the changes made (and the steps taken) clearly, since this is a structure-based work and all the issues raised were related to the structure. Given these discrepancies in the statistics, I am not convinced that the data processing for the structure was done properly.

Author Response

Our thanks to the reviewer for his/her comments which have improved the manuscript.

I can understand the referee's concern about the increase in resolution - this resulted from a re-processing of all the available data and he/she is correct to be concerned about a significant change in the resolution of the data presented. We diffracted a number of crystals (ranging from 3.4 - 2.04A) during the course of this project and here report the "best" we have available after re-examining all available datasets. 

Should the referee wish to see the primary data and processing for the improved crystal data here reported I have uploaded all the original XDS processing log files, as well as the first 50 of the 600 frames to:

https://drive.google.com/open?id=1QeUt1eHyF1-7Ro2ul2z3iyQzim8pm4n-

I hope he/she can convince herself of the quality of the diffraction from these images, but I am happy to upload the remaining 550 if requested.

An abstraction from the CORRECT.LP is also attached below (full file is available from the download link above). Please note that data was ultimately scaled with SCALA and truncated at 2.15A, resulting in slightly different statistics as presented in our manuscript.

SUBSET OF INTENSITY DATA WITH SIGNAL/NOISE >= -3.0 AS FUNCTION OF RESOLUTION
RESOLUTION NUMBER OF REFLECTIONS COMPLETENESS R-FACTOR R-FACTOR COMPARED I/SIGMA R-meas CC(1/2) Anomal SigAno Nano
LIMIT OBSERVED UNIQUE POSSIBLE OF DATA observed expected Corr
5.84 3938 1606 1712 93.8% 1.4% 1.7% 3909 48.38 1.8% 99.9* 13 0.723 213
4.23 6493 2579 2740 94.1% 1.8% 1.8% 6450 43.65 2.4% 99.9* -16 0.757 294
3.48 8192 3250 3449 94.2% 2.4% 2.2% 8143 33.71 3.1% 99.9* -10 0.783 281
3.03 10015 3921 4055 96.7% 3.8% 3.9% 9942 20.54 4.9% 99.8* -6 0.772 299
2.71 11136 4347 4552 95.5% 8.2% 8.2% 11049 10.90 10.4% 99.4* -8 0.618 287
2.48 12419 4841 5033 96.2% 14.5% 14.6% 12325 6.65 18.6% 98.4* -8 0.644 277
2.30 13367 5214 5432 96.0% 24.2% 24.6% 13255 4.17 30.9% 96.2* -9 0.657 237
2.15 14441 5640 5859 96.3% 37.0% 37.2% 14312 2.81 47.3% 90.4* 1 0.751 233
2.03 14956 5884 6193 95.0% 51.7% 52.0% 14770 2.02 66.1% 76.0* -8 0.667 187
total 94957 37282 39025 95.5% 4.2% 4.2% 94155 13.66 5.3% 99.9* -6 0.710 2308

Reviewer 2 Report

Authors have addressed my concerns.  I have not seen the updated validation report, but it appears that re-processed data is at higher resolution which should result in improved model quality. 

Author Response

Many thanks to the reviewer for his/her help in improving the manuscript

Reviewer 3 Report

I had a view on the updated version and the crystallography statistics look fine now.
However, the updated refinement procedure should be detailed in the method section.
Is the PDB entry 6SU9 correct? I‘m still not quite happy with the discussion as it should include more comparing literature.

Author Response

Our thanks to the review for his/her comments to improve the manuscript.

Detailed answers to his/her comments are provided below:

We have updated the refinement section in the manuscript  The PDB entry 6SU9 is correct and replaced the original lower resolution entry We have expanded the discussion 

Round 3

Reviewer 1 Report

The revised draft of the manuscript looks better with minor edits. The statistics for the crystal structure look fine to me, although the authors have not answered my queries about the strange similarities between the two statistics, clearly and to the point. In addition to my question on the drastic increase in resolution, I have also raised concerns about why some of the other statistics (R-merge and I/sigma (I)) between the two datasets remained same despite significant reprocessing and/or inclusion of newer data. I do appreciate the willingness of the authors to share the raw images and other details. However, it would serve the authors better if they can answer the reviewers' questions specifically and point-by-point with all the details they can provide, rather than making vague statements.

For example, instead of mentioning 'increase in resolution resulted from reprocessing of all the available data', they could have been more specific about:

What different reprocessing steps were done?

Did they include new (2.1 A ) data from other crystals which were not previously analyzed or did they exclude some of their previous noisy images/data from the processing?

Why did they not select their 'best' diffraction (2.1 A) data in the first place, instead of processing a lower 2.5 A resolution datasets?

Minor edits:

Figure 3: Consider removing the broken red lines and the arrows23. Instead use labels a-d for the individual figure panels and describe them in the figure legend. Figure 3: Change Mg ion to Mg2+ ion.

Author Response

Referee report: 

The revised draft of the manuscript looks better with minor edits. The statistics for the crystal structure look fine to me, although the authors have not answered my queries about the strange similarities between the two statistics, clearly and to the point. In addition to my question on the drastic increase in resolution, I have also raised concerns about why some of the other statistics (R-merge and I/sigma (I)) between the two datasets remained same despite significant reprocessing and/or inclusion of newer data. I do appreciate the willingness of the authors to share the raw images and other details. However, it would serve the authors better if they can answer the reviewers' questions specifically and point-by-point with all the details they can provide, rather than making vague statements.

For example, instead of mentioning 'increase in resolution resulted from reprocessing of all the available data', they could have been more specific about:

What different reprocessing steps were done?

Did they include new (2.1 A ) data from other crystals which were not previously analyzed or did they exclude some of their previous noisy images/data from the processing?

Why did they not select their 'best' diffraction (2.1 A) data in the first place, instead of processing a lower 2.5 A resolution datasets?

Author Response: 

Our apologies for the vague reply. We had hoped that providing access to the raw data and original processing files would provide a clear answer to the source of the statistics quoted in Table 1 and address the referee's concern that "I am not convinced that the data processing for the structure was done properly."

1. The data reported here comes from a single crystal processed in the same manner that all diffraction data in the project were - using XDS as shown in the files uploaded to the google drive and as stated in the manuscript. Our answer in the previous rebuttal would have been better phrased as: we report the "best" crystal we have available after re-examining all available datasets.

2. We have provided the original data (50/600 frames) to allow the referee to judge for him/herself if the data quality appears consistent with Table 1 (with an offer to supply the rest upon request) as well as the original CORRECT.LP of the entire 600 frame that show statistics as reported in Table 1 originates from a single crystal processed in a single run.

This shown by CORRECT.LP (available on google drive) line 13 & 14:

NAME_TEMPLATE_OF_DATA_FRAMES=../PdxK_P_3_???.mccd TIFF
DATA_RANGE= 1 600

3. The best diffracting crystal reported here was not selected in the first place, due to a confusion over which was the best data set to use to solve the structure (see below). However, the statistics reported in the table do correspond to the 2.15A dataset now reported in the revised manuscript, as demonstrated in the uploaded data and processing files. We note that both the latest Table 1 values and those of the previous submission are in agreement with our previously published values in Kronenberger et al 2014 Acta Cryst F70 1550-1555.

4. To recheck the processing we have performed it again today and inserted these latest values into Table 1. The CORRECT.LP with this latest reprocessing is also uploaded to the google drive as CORRECT_Oct10_2019.LP in the XDS subdirectory. 

The reason for this confusion is all rather mundane. During the data collection I keep notes on paper of the processing results as further crystals are collected - essentially the statistics reported in Table 1 (I/sigI /R/CC/etc). As this project required multiple datasets I was collecting over the full 24 hours available. Apparently, when I'm tired, my handwritten "2.1" looked like "3.1" to my student (the lead author) when I passed over the data and the accompanying handwritten notes. He subsequently used the next best data - where he could clearly read "2.5" - and solved the structure (which unsurprisingly is almost identical to that at 2.1). Later (when preparing this manuscript) I was able to more accurately read my own writing when Kai asked me to enter the details into Table 1 and inserted the values for the 2.1A data set - rather than those associated with the 2.5A dataset he'd used to solve the structure - hence the confusion on the table entries and the essentially unchanged numbers. My error was not spotting that a lower resolution data set had been used to solve the structure. 

Thankfully, upon reexamination of all data sets collected (based on the questions from this referee and others), we spotted this mistake and re-processed and re-refined the structure against the significantly better dataset  (although due to the transcribing error above, the processing statistics remain essentially unchanged). Without the referees' dilligence we likely would not have been able to identify and correct this error in the previous revision, for which we are extremely grateful.

Due to this mix up of us using the "wrong" dataset to successfully solve the structure (as well as submitting a manuscript with incorrect processing statistics), I decided to take the highly unusual step of providing the raw data and processing log files to ensure transparency as well as performing the refinement against the new dataset as reported in the first revision. This was done to address the referee's concern that "I am not convinced that the data processing for the structure was done properly." Again, should the referee wish to download the entire dataset to confirm for him/herself the values reported in Table 1, I'm happy to provide it.

While this may be a vaguely amusing story I hope the referee doesn't think we need to modify the manuscript to include it. 

Minor edits:

Figure 3: Consider removing the broken red lines and the arrows23. Instead use labels a-d for the individual figure panels and describe them in the figure legend. Figure 3: Change Mg ion to Mg2+ ion. 

These changes have been made.

Reviewer 3 Report

The authors have significantly improved their presented structure by a more rigorous refinement procedure. I am happy with publication of the manuscript. There are only some minor issues to address. 

Figure 3 lacks the respective contour levels. Why is the omit map not shown in the lower left panel?

Line 87: "...from 3.4A to 2A" 

In line 96 you state that at least 98% of all residues are found in the favored region. However, table 1 only shows 95%. This should be corrected. 

Line 279: "...wild-type.."

Author Response

The authors have significantly improved their presented structure by a more rigorous refinement procedure. I am happy with publication of the manuscript. There are only some minor issues to address. 

Figure 3 lacks the respective contour levels. Why is the omit map not shown in the lower left panel?

The figure shown is indeed the omit map. We have changed the legend accordingly.

Line 87: "...from 3.4A to 2A" 

In line 96 you state that at least 98% of all residues are found in the favored region. However, table 1 only shows 95%. This should be corrected. 

Line 279: "...wild-type.."

These errors have been corrected